# How Living in Vulnerable Conditions Undermines Cognitive Development: Evidence from the Pediatric Population of Guatemala

**DOI:** 10.3390/children8020090

**Published:** 2021-01-29

**Authors:** Joaquín A. Ibáñez-Alfonso, Rosalba Company-Córdoba, Claudia García de la Cadena, Antonio Sianes, Ian Craig Simpson

**Affiliations:** 1Department of Psychology, Human Neuroscience Lab, Universidad Loyola Andalucía, 41704 Sevilla, Spain; jaibanez@uloyola.es (J.A.I.-A.); rcompany@uloyola.es (R.C.-C.); icsimpson@uloyola.es (I.C.S.); 2ETEA Foundation, Development Institute of Universidad Loyola Andalucía, 14004 Córdoba, Spain; 3Department of Neuropsychology, Universidad del Valle de Guatemala, Guatemala 01015, Guatemala; claudigd@uvg.edu.gt; 4Research Institute on Policies for Social Transformation, Universidad Loyola Andalucía, 14004 Córdoba, Spain

**Keywords:** children, adolescents, cognitive performance, assessment, neuropsychology, vulnerability, violence exposure, poverty, 2030 agenda

## Abstract

Low-socioeconomic backgrounds represent a risk factor for children’s cognitive development and well-being. Evidence from many studies highlights that cognitive processes may be adversely affected by vulnerable contexts. The aim of this study was to determine if living in vulnerable conditions affects childhood cognitive development. To achieve this, we assessed the performance of a sample of 347 Guatemalan children and adolescents aged from 6 to 17 years (*M* = 10.8, *SD* = 3) in a series of 10 neuropsychological tasks recently standardized for the pediatric population of this country. Two-fifths of the sample (41.5%) could be considered to have vulnerable backgrounds, coming from families with low-socioeconomic status or having had a high exposure to violence. As expected, results showed lower scores in language and attention for the vulnerable group. However, contrary to expectations, consistent systematic differences were not found in the executive function tasks. Vulnerable children obtained lower scores in cognitive flexibility compared to the non-vulnerable group, but higher scores in inhibition and problem-solving tasks. These results suggest the importance of developing pediatric standards of cognitive performance that take environmental vulnerable conditions into consideration. These findings, one of the first obtained in the Guatemalan population, also provide relevant information for specific educational interventions and public health policies which will enhance vulnerable children and adolescent cognitive development.

## 1. Introduction

Half of the world’s at-risk population due to multidimensional poverty are children [1]. Living in vulnerable conditions affects these children’s future opportunities from very early years by lessening their developmental potential. Perhaps surprisingly, childhood poverty is not only a problem for children in developing countries. An important number of children in developed countries live with insufficient resources to guarantee their proper physical, psychological, and social development. One consequence of poverty is malnutrition, which prevents the adequate development of one in three children around the world [2]. Given the global repercussions of poverty and its associated negative consequences, the United Nations (U.N.) has initiated diverse actions to combat this problem including adopting the 2030 Agenda for Sustainable Development [3]. The 2030 Agenda contains 17 Sustainable Development Goals (SDGs), including such objectives as eradicating poverty and hunger, and to improve the quality of education globally [4]. By proposing the involvement of governments in the implementation of these initiatives, the idea is to ensure sufficient economic and social resources are made available to fulfill those goals. Hence, the SDGs represent an opportunity to improve the lives of these children and to provide equal life opportunities. To achieve the SDGs, a shift in the way academic research is undertaken is needed due to the fact that the current assessment and reward systems used to evaluate academic performance and research worth often do not take into account the importance of transformative research designed to improve societal conditions [5]. Such a shift could lead to the creation and implementation of a whole new set of policies and interventions which would allow the scientific community to actively participate in achieving the SDGs [6]. Our study embraces this commitment by studying how vulnerable contexts have implications for children’s development.

### 1.1. Cognitive Development Under Vulnerable Conditions

Johnson, Riis, and Noble [7] conducted an extensive literature review of the relationship between socioeconomic status and brain development, noting that an important number of studies have taken into account the implication that a lack of resources and exposure to toxins and stress have for a child’s brain function and structure. Special attention needs to be taken regarding the effect of exposure to stress in early years. Stress is usually defined as exposure to life events that requires a subsequent adaptive behavior [8]. Previous studies have described how stressors caused by poverty affect children and adolescent health status [9], with some authors describing these effects as disturbances in brain development [10,11]. These consequences are related to the activation and reaction of the Hypothalamic-Pituitary-Adrenal axis. This results in increased cortisol levels, which especially affect prefrontal areas that are involved in cognitive functions such as working memory, cognitive flexibility, and inhibition, along with more general functions, such as self-regulation cognitive and emotional processes. Those factors end up negatively affecting children’s health and psychological wellbeing as well as quality of life [12]. In turn, this results in difficulties for children in escaping the effects of these early negative influences, even once adulthood has been reached [13]. Given these relationships, it is unsurprising that diverse studies have shown the alterations in subcortical areas such as the hippocampus [14,15,16,17], the amygdala [14,16], along with other dorsolateral prefrontal cortex volumes [15] in children from low socioeconomic status (SES) backgrounds. These alterations can impact cognitive performance in a number of areas, including attention, memory, motor skills, language, and executive functions.

Attention allows the individual to be oriented towards and focused on a specific target. SES seems to be related with performance in attentional tasks. For example, a study that compared attention tasks in a group of 10–13-year-old (*n* = 141) children, and found that children from vulnerable backgrounds showed less attentional efficacy than children from nonvulnerable backgrounds [18]. One possible explanation for this finding is that children who live in chaotic households where nothing is predictable, become adapted to this context and consequently show less attentional selectivity [19]. The adaption to unpredictability could be a strategy to cope with the limited sense of control that they can experience on a daily basis. What are the consequences for low SES children whose attention is negatively impacted by their circumstances? Using selective attention properly allows the individual to be oriented to a specific goal in a given task. This capacity is important in early ages and is related with school performance, since without a proper control of this kind of attention, children could face problems when ignoring irrelevant items [20]. In this sense, low-SES children are inclined to attend similarly to targets and irrelevant stimuli, which leads to interference in their attentional performance compared with their counterparts [21]. Mezzacappa [22] evaluated the attentional capacity in children of 6 years of age from diverse socioeconomic backgrounds, finding that low-SES children tend to show less speed and accuracy in measures of attention. Taking this into account, the results of research focusing on attention suggest that economically disadvantaged children have greater problems in inhibiting irrelevant information in comparison with children from wealthier backgrounds [18,23,24]. Despite the probable negative consequences low SES has for attention in young children, there is some evidence that they may be able overcome these disadvantages at a later age. In their study, Lupien et al. [25], found that low-SES children at 6 years of age performed worse than their peers in selective attention tasks, but this tended to change with age as no significant differences were found between low-SES adolescents and their peers. Nevertheless, taking all of these results into account, the results suggest that economically disadvantaged children show a lower performance in attention-related tasks.

Memory is an important cognitive domain since it allows the individual to retain internal and external information and learn from experience. Previous studies suggest that low SES affects a variety of memory-related processes such as visual and verbal memory, in both episodic and semantic systems, up until the age of 9 [26]. Farah et al. [27] found that children from low-SES contexts performed significantly worse than matched controls in incidental learning tasks. These authors suggested that the results were due to the effects that stress causes in memory systems. In a study conducted with adolescents [28], procedural memory did not show significant relations with SES levels. Nevertheless, in the same study, higher socioeconomic indicators were related with better performances on working memory tasks [28].

In one study, using the type of school as an indicator of SES, children from private schools showed better performance in memory tasks of visual encoding and deferred evocation [29]. Burneo-Garcés et al. [30], studied verbal and visual memory performance using immediate recall, delayed recall, and recognition tasks. These authors found that children from medium SES backgrounds performed better than their lower SES background peers on verbal memory tasks that measure immediate and delayed recall as well as on recognition tasks. The medium SES children also performed better in visual memory tasks involving immediate recall, although there was no difference between the two groups in delayed recall or recognition tasks.

Motor skill refers to the ability to coordinate the movement required to complete an action. Some studies have examined the relationship between SES and components related with motor development. For example, in a study conducted with school-age children, SES was shown to have an impact on motor development scores [31]. Adding as a mediation variable the adequacy of household environment, assessed by the Home Observation for Measurement of the Environment (HOME) inventory [32], these authors reported that the proportion of variance in motor development explained by SES increased from 9 to 13%. Other studies in which children who attended public and private schools were compared have shown that children attending public schools obtained lower scores in fine motor skills in comparison with their private school counterparts [33,34].

Language is the ability that allows an individual to communicate with others. The relationship between SES and children’s verbal language ability has been evaluated in several studies, with negative effects of socioeconomic status generally being reported [27,35,36]. Differences have also been found in young children’s writing skills [26] and vocabulary [37]. Some aspects of the home environment, such as the availability of resources in the home, as well as parental educational level, can impact receptive vocabulary in children at different ages [38]. Lack of resources and low parental education are often associated with low SES levels and children from low SES settings are, among other factors, exposed to less complex oral language [39], which could in turn affect performance in language-related tasks. As Fernald, Marchman, and Weisleder showed in their study [40], there are significant differences in language aspects such as processing efficiency and vocabulary learning in infants at 18 months of age depending on their socioeconomic status. A similar study from Ecuador [30] compared performance in verbal comprehension and verbal fluency tasks, finding that the medium SES group performed better than the lower SES group. In a study of older children, adolescents from diverse socioeconomic backgrounds exhibited lower scores in language-related tasks, especially in vocabulary tests [41]. Farah et al. [27] examined neurocognitive functioning using a battery of tests defined jointly by functional and anatomical criteria. For the language-based tasks, SES disparities were found in the perisylvian system, an area of the brain known to be involved in language processing. Thus, a neurological correlate of these SES related differences in language tasks could be supported by existing differences in the left perisylvian area.

Executive functions (EF) are known as a compendium of processes which allow the self-regulation of goal-directed behaviors. Due to their importance, these processes have been well described and studied in the literature [42]. Evidence has been found linking socioeconomic status and measures of intelligence [26,43], EF in general [44], and specific processes such as working memory and cognitive control [27,36] with children at social risk obtaining lower scores in EF-related tasks [27,45,46,47]. A recent study highlighted family functioning as a source that could explain the relation between SES and the difficulties found in executive function performance in children [48]. EF are composed by a cognitive process that allow the individual to solve new and complex tasks, therefore the significance of EF in school performance is evident, being a factor that has been shown to be a mediator between poverty and school achievement in a multiracial study [49] and in a study conducted with a sample of vulnerable children from Argentina [50]. Working memory (WM) is one of the EF cognitive processes most studied and is related to the ability to operate with short-term information. This process is commonly understood to be divided into the visuospatial sketchpad, episodic buffer, and the phonological loop. In one study, when comparing low SES children from urban and rural areas with their wealthier counterparts, Tine [51] observed that urban low SES children performed worse than wealthier children in visuospatial and phonological tasks, while low SES children from a rural area underperformed in visuospatial WM. Leonard et al. [28], assessed WM among other domains in economically disadvantaged adolescents and their peers, finding that lower SES negatively affects this domain. Moreover, experiencing poverty during childhood, impacts the visuospatial WM component in adults [52].

### 1.2. Guatemalan Context

Based on this previous evidence, it seems that growing up in a low-SES environment potentially creates negative impacts for these children from an early stage of development. Specifically, the lack of resources available to these children gives rise to difficulties in areas such as physical and psychological health, and academic success, among others. However, the majority of studies that have examined this issue have been undertaken in high-income countries, especially in the United States [24,51,53]. Consequently, there is a significant lack of similar studies that have been carried out in middle-to-low-income countries. In order to address this shortfall, in this study we assessed cognitive skills such as motor skills, memory, attention, language, and executive functions in a group children and adolescents from Guatemala.

Within the countries of the Central American isthmus, Guatemala is the largest country with approximately 13 million inhabitants. It has a complex history in which 25 cultural ethnic groups have converged, 22 of which are Mayans, along with one Xinca group, one Garífuna group, and one non-indigenous group. Thus, the country is extraordinarily influenced by the indigenous culture to which 43.8% of population belongs [54]. All of these different ethnic groups have to face many difficulties on a daily basis, such as discrimination and racism. Additionally, despite being the largest economy in the Central American region, with regards to the indicators of nutrition, health, violence exposure, education, employment, and the well-being of the population, there are huge gaps as many of these factors have not been adequately studied in this region. A significant source of vulnerability for Guatemalan children and adolescents are those factors related with the familiar socioeconomic status. Importantly, what is considered to be medium-low SES in Guatemala, would be seen as low SES or indications of a vulnerable population in countries with a larger economy and extended welfare. Hence, the proportion of children in Guatemala living in what would be considered low SES conditions in other countries is almost certainly higher than indicated by the official figures.

Despite the fact that the government of Guatemala has included factors such as access to education and poverty reduction as a priority in its National Development Plan [55], there is still a long way to go. Guatemalan children and adolescents experience different types of vulnerability depending on the area in which they reside. Being aware of the realities these children face in their daily lives can help in understanding how the different forms of vulnerability interfere with their proper psychological and cognitive development.

A study conducted by the Food and Agriculture Organization of the United Nations found that in 2014 around 25% of the rural population of Guatemala were living in conditions of extreme poverty, with a further 50% of the rural population were considered as suffering from poverty [56]. This fact means that rural children and adolescents find themselves in a vulnerable position from an early age. Malnutrition is a condition associated with poverty. Consequently, long term nutritional programs have been implemented in rural areas of Guatemala for children due to the importance of adequate nutrition in cognitive development [57,58]. Moreover, although children and adolescents from urban areas may suffer less from malnutrition, they nevertheless suffer from other hazards such as being witness to, or a victim of extreme violence, a factor which has also been shown to negatively affect children’s development. For example, witnessing intimate partner violence (defined as parental partner violence of a physical, emotional, or psychological nature [59]) in different stages of childhood and adolescence has been shown to have a negative effect on cognitive development [60]. Moreover, suffering from violence has been shown to have an impact on brain functions in a sample of Latin American preadolescents [61]. This sample showed higher cortisol levels and less activation of the right hemisphere areas involved in social perception as well as cognition. A more recent study examined the effects of exposure to violence along with food concern on the relationship between socioeconomic characteristics and mental health indicators in children and adolescents living in vulnerable conditions in Guatemala [12]. These authors found a moderating effect of exposure to violence on the relationship between socioeconomic characteristics and depression, anxiety, and health-related quality of life such that higher exposure to violence resulted in higher scores in depression and anxiety indicators. Moreover, in the same study adolescents showed higher depression and anxiety scores and lower health-related quality of life scores compared to children suggesting that these mental problems increase when children grow older. Food insecurity experienced by parents was not found to be a moderating variable between socioeconomic and mental health variables.

### 1.3. Aim of Study

The objective of the present study was to determine if living in vulnerable conditions negatively impacts the cognitive development of children and adolescents. To accomplish this, we compared the cognitive performance of vulnerable and non-vulnerable children living in Guatemala. To achieve a broad comparison, 10 different neuropsychological tests were chosen to encompass the most relevant cognitive processes: Motor skills, memory, attention, language, and executive functions. A second motivation for choosing these specific tests was that they had recently been standardized for the Guatemalan population. We hypothesized that there would be significant differences between the two groups, even after applying the same normative inclusion criteria. Lower scores for the vulnerable group were expected in all cognitive domains assessed.

## 2. Materials and Methods

### 2.1. Participants

A total of 397 participants underwent a screening process for inclusion in the study. Inclusion criteria were: (a) Having Spanish as mother language or proficient as a second language, (b) having an Intelligence Quotient of ≥80, (c) scoring below the threshold scores in depression and anxiety questionnaires, (d) not having previous history of neurological or psychiatric disorders, and (e) not having a history of drug use. Based on these criteria, 50 participants were excluded from the study (42 from the vulnerable group, 8 from the non-vulnerable group). The inclusion criteria were chosen based on our research goal. We aimed to assess healthy children (for which we used the questionnaires) with no intellectual or mental health disturbances (for which we used the TONI-2, anxiety and depression questionnaires, respectively). Since our goal was to assess a pediatric population, we selected the age range from 6 to 17, taking 6 as the minimum age because we consider that from this age, children are able to complete the vast majority of neuropsychological tasks used in the study.

The vulnerable group was characterized by living in areas where extreme poverty and violence exposure were key difficulties in their lives. The non-vulnerable group was composed of participants from medium-low SES backgrounds whose environment was not especially characterized by these difficulties. In this regard, it is important to consider that our vulnerable and non-vulnerable groups were comprised of children from both low and medium-low SES backgrounds. Accordingly, in this study we did not set out to compare high-SES with low-SES participants. The final sample was composed of 347 Guatemalan children and adolescents aged from 6 to 17 years (*M* = 10.8, *SD* = 3.0). Within the whole sample, the vulnerable subsample was composed of 144 participants (71 girls, 49.3%) recruited from three schools of the educative institution Fe y Alegria (age *M* = 11.0, *SD* = 3.7). Two of the schools were based in rural areas of the Totonicapán department, which is located in the southwest of the country (91 participants, 63.2% of the vulnerable sample) while the third school was based in the suburban area of Guatemala City, in the district of Villanueva. Within the vulnerable sample, children recruited from the rural schools suffered from poverty conditions while children recruited from the urban school were routinely exposed to violence on a daily basis since the school was located in an area bordered by two competing criminal gangs. Thus, these schools were chosen because of the especially vulnerable conditions faced by students that experience a high exposure to violence and poverty. Within the vulnerable group, 75 rural participants (52.1% of the vulnerable group, 21.6% of the total sample) were bilinguals (Spanish-K’iche). The non-vulnerable group was composed of 203 participants (95 girls, 46.8%; age *M* = 10.7, *SD* = 2.5). These participants came from one private and one public school located in the capital, Guatemala City, where the children’s families had a medium socioeconomic status. The mean level of parental education (MLPE), calculated using the average number of years that the primary caregivers went to school, was 6.5 (*SD* = 4.5) for the vulnerable group, and 10.5 (*SD* = 4.1) for the non-vulnerable group. All participants gave their informed consent prior to participating in the study.

### 2.2. Instruments

The questionnaires and screening tests used to ensure that participants met our inclusion criteria for physical and mental health were the same as those used in a previous study conducted by Rivera & Arango-Lasprilla [62]. The 10 neuropsychological tasks used to assess the sample were chosen since they encompass the measurement of the most common cognitive processes: Motor skills, memory, attention, language, and executive functions. These neuropsychological tasks are included in Appendix A. Normative data for the Spanish-speaking pediatric population is available for all the following neuropsychological tests, including specific normative data for the Guatemalan pediatric population [63,64,65,66,67,68,69,70,71,72].

### 2.3. Procedure

In order to inform families about the study goals and procedure, the researchers met teachers and parents in each school. Parents were then asked if they would allow their children to participate and to sign the informed consent, as well as complete the sociodemographic, clinical, and language questionnaires. Children with consent from both groups were then pseudo-randomly selected for participation in the study as we implemented a constraint of trying to balance age and gender across the vulnerable and non-vulnerable groups.

Children were individually assessed for approximately 120 min in a well-illuminated room provided by each school. Children received a rest break when one hour of assessment was completed. The assessments were performed by trained psychologists. The screening tasks were administered in the same session and immediately corrected in order to determine if the participant met the inclusion criteria. Each neuropsychological test from the battery was administered in a different order, randomly assigned to each child. Some tasks could not be completed by some children because of different reasons: The child was unable to read, the numerical order was not internalized, or the participant refused to complete the task.

### 2.4. Data Analysis

Statistical analyses were conducted using SPSS version 26 [73]. First, missing values were coded to indicate the reason for the lack of information. Subsequently, outliers in the raw scores were modified to have a value of ±3 *SD* using the winsorinzing method [74].

All comparisons were done with bootstrapping using 95% confidence intervals and 1000 repetitions in order to control for possible bias due to the non-normality present in some of the distributions. Although the vulnerable group were slightly older (0.3 years, 95%CI −0.4, 1.0), this difference was not significant (*t*(235.5) < 1, *p* = 0.400). In contrast, the 4 year difference (95%CI −4.9, −3.1) in the mean level of parental education favoring the non-vulnerable group was significant (*t*(345) *=* −8.62, *p* < 0.0001). Chi square test revealed that gender distribution between groups was not significant, χ^2^ (1) = 0.212, *p* = 0.645.

Means and standards deviations were analyzed in all neuropsychological tasks. A series of independent samples *t*-test was conducted to evaluate the differences in scores between the vulnerable and non-vulnerable groups in the 10 neuropsychological tasks.

## 3. Results

The raw scores obtained by the vulnerable and non-vulnerable groups in the 10 neuropsychological tests along with the results of the independent samples *t*-test comparisons are described in Table 1, Table 2, Table 3 and Table 4. The raw scores obtained by age group and vulnerable condition are included in the Table 5.

The comparisons are shown in Table 1, Table 2, Table 3 and Table 4, ordered by the cognitive domain. For the motor skill and memory tasks, although all of the means were slightly higher for the non-vulnerable group, none of the differences were significant and all effect sizes were small (Cohen’s *ds* ≤ 0.15). Nonetheless, significant differences between the groups were found in other cognitive processes. With regards to attention, the non-vulnerable group performed better on both tasks. For the D2 concentration index, the size of the effect was medium-to-large (*p* = 0.001, *d* = 0.61) whereas for the SDMT (Symbol Digit Test) task, the effect was small-to-medium (*p* = 0.012, *d* = 0.31). Regarding language performance, the non-vulnerable group scored significantly higher in the majority of tests, with the largest effect sizes being for receptive vocabulary PPVT-III (*p* = 0.001, *d* = 0.87), and the language comprehension Token test (*p* = 0.001, *d* = 0.63). Measures of language expression also showed a general better performance by the non-vulnerable group. In the phonological verbal fluency series all differences were significant (*ps* < 0.05, Cohen’s *ds* between 0.22 and 0.38). For semantic fluency, the non-vulnerable group was superior in the animal category (*p* = 0.016, *d* = 0.29), but surprisingly, not in the fruit category (*p* = 0.922, *d* = 0.03). Results in executive function tasks were heterogeneous. In the TMT (Trail Making Test) task the vulnerable group needed significantly more time to complete both the A and the B series (*ps* = 0.001, *ds* > 0.75), which may be interpreted as a slower processing speed (A) and worse cognitive flexibility (B). In contrast, in the interference score of the Stroop task, which is related to the cognitive inhibition ability, the vulnerable group showed better performance (*p* = 0.001, *d* = 0.44), without significate differences in some automatic processes such as color naming (*p* = 0.506, *d* = 0.09). In the case of the M-WCST (Modified Wisconsin Card Sorting Test) task, vulnerable participants also exhibited better performance in terms of problem-solving ability (correct categories, *p* = 0.006, *d* = 0.33), although no difference in the flexibility measure was observed (perseverative errors, *p* = 0.765, *d* = 0.02).

## 4. Discussion

The purpose of this study was to determine if living in vulnerable conditions negatively impacts cognitive development. To achieve this, we used 10 neuropsychological tests recently standardized for the Guatemalan pediatric population to evaluate a broad range of cognitive abilities in two groups of children and adolescents. These 10 tests were chosen because they encompass the measurement of the most relevant cognitive processes: Motor skills, memory, attention, language, and executive functions. The vulnerable group was composed of children who are living in poverty conditions and/or are frequently exposed to violence in their daily lives. The non-vulnerable group was composed of children from medium-low SES backgrounds who rarely experience violent incidents or suffer extreme poverty in their daily lives.

As per our expectations, the vulnerable group underperformed in language- and attention-related tasks in comparison with their non-vulnerable counterparts. In contrast, results for the executive function tasks differed in some cases from those expected.

Regarding motor skills, no significant difference was found between the vulnerable and non-vulnerable groups on the ROCF (Rey-Osterrieth Complex Figure Test) copy subtest. This result differs from previous studies which did report differences [33,34,75]. The differences between the results reported here and what has been found in previous studies could be due to the different way in which children are classified in terms of vulnerability. Previous studies have used the type of school (public vs. private) to classify participants [34,75], whereas in this study we used the socioeconomic characteristics of the area in which the schools were based in order to classify the children. Potentially, the different methods used to form the vulnerable and non-vulnerable groups mean that the results of the different studies are not directly comparable.

No differences were found when comparing vulnerable and non-vulnerable groups in terms of memory performance in the ROCF memory subtest and in verbal memory subtests. This result does not agree with previous findings as the economically disadvantaged group was expected to obtain lower scores [30]. A previous study which examined how early ages’ Socioeconomic Context (SEC) affects cognitive functions throughout an entire life span [76], showed that higher SECs in early ages and adulthood were related to better scores in verbal memory recognition. The literature regarding memory of participants with a background of poverty shows a link between lower family income and the hypo-reactivity of the cortisol response, which has been shown to be linked to associative memory in children [77]. The lack of participants from high-SES backgrounds, being our whole sample from medium-low and low SES, may have reduced the expected effects.

The vulnerable group in the present study showed poorer performance in attentional tasks in comparison with the non-vulnerable group, although the effect size was small to medium (SDMT, *d* = 0.31; D2 concentration index, *d* = 0.61). This is in accordance with previously reported findings which have also found differences [18,19,47]. This pattern of results has been replicated in sustained attention tasks involving infant participants [24,78]. In fact, low-SES has recently related to a higher risk of suffering from symptoms of attention deficit/hyperactivity disorder (ADHD) [79]. In the D2 test, the concentration index offers information about the child’s ability to maintain attention and response accuracy for an extended time period. Thus, this index provides information about how selective and sustained attention function. The way in which contextual characteristics influence basic cognitive processes that are attention-related has been studied [22], and are in the line with the results reported here. Furthermore, the present results agree with those found in studies which have evaluated Event-related Potentials (ERPs) and Electroencephalographic (EEG) measures, since there seems to exist a clear neural correlate between SES and attention-related tasks [23,80]. This means that there is evidence of the neurophysiological substrate of the implications of SES differences in terms of attentional performance. Our results highlight the profound negative impact that attention development may suffer under vulnerable conditions, such as violence exposure and extreme poverty. The relation between attention and socioeconomic disadvantage has been of such importance that some authors have proposed specific interventions to diminish this developmental gap [81].

Regarding language, vulnerable children and adolescents achieved a lower performance than their non-vulnerable counterparts, in the majority of language tasks. Other authors have also observed a similar impact of SES on language [27,47,82]. Our results for receptive vocabulary, as assessed using the PPVT-III, agree with previous studies which have reported lower scores in the case of low-SES children on different language measures such as semantic processing [36], receptive vocabulary [38], language comprehension tasks [30,41], and fluency [83]. The receptive vocabulary results in the present study warrant further comment. The difference between means of 23 points favoring the non-vulnerable group is a large effect (Cohen’s *d* = 0.87). When interpreting the data, it is important to consider that the vulnerable group had two factors that could be related with their low scores in language tasks: Vulnerability and the large percentage of bilinguals included in the sample. The way in which the bilingualism impacts language tasks has been studied, and in general, bilingualism tends to be associated with reduced language scores in children (for a review, see Bialystok) [84]. Moreover, there are studies that have considered the impact of both bilingualism and a low-SES background, finding that each factor independently has a negative impact on children’s linguistic performance [85]. Furthermore, language performance has been studied in indigenous minorities in comparison with the non-indigenous population in a sample of Australian school students, with lower scores in receptive vocabulary obtained by the indigenous group [86]. Thus, it plausible that in cases where vulnerability and minority languages converge, the development of the majority language is especially at risk.

One curious result from the language tasks was the conflicting findings in the two semantic fluency tasks. While the non-vulnerable children generated more words in the animal category (*p* = 0.016, *d* = 0.29), there was a null effect in the fruit category (*p* = 0.922, *d* = 0.03). The first point to note here is that the mean difference in the animal category was small, being just 1.3 words, so in a practical sense, the two groups are not too dissimilar. Nevertheless, we offer the following tentative explanation for the different pattern of results. These two tasks were carried out in Spanish. As previously noted, there were a large number of bilinguals in the vulnerable group. Given the importance of food in general to the lives of these children and their families, it is possible that the bilingual children have learned the names for fruit in both their native language and Spanish and were thus able to generate as many fruit names as their non-vulnerable counterparts. In contrast, the children in the vulnerable group may not have been exposed to as many non-native Guatemalan animals (via reading, personal experience, parental tuition, etc.) as their non-vulnerable counterparts, and such were not as capable of generating as large a range of animal names in the Spanish language as the children in the non-vulnerable group.

In regard to executive functioning, results in this study contrast in some cases with previous literature that links vulnerability conditions with worse EF [27,36,45,87]. Although our results in the TMT were congruent with previous evidence, the interpretation of results obtained in the Stroop test and in the M-WCST is especially challenging. For example, a previous study conducted in Ecuador with medium and low-SES participants focusing on cognitive inhibition tasks found that performance in resistance to interference was better for vulnerable children [30]. Although this result was the opposite of that expected by these authors, it is congruent with the results reported here. We offer the following account as a possible explanation for these results. The use of this test is recommended for children from 6 years old because simple word reading skills should already be achieved by this age. However, this does not occur in all contexts. School milestones are reached at different ages depending on the country and the characteristics of the study population. In this case, the fact that children from vulnerable backgrounds may not be as proficient in reading as their non-vulnerable counterparts could be reducing the word-color reading interference effect. Thus, a possible explanation for this result may be that an interference effect is not present in these children because their low levels of literacy do not give rise to such an effect using this literacy-based task. This account may also explain why an almost significant difference was found in the word naming task (*p* = 0.089, *d* = 0.22, mean difference of 4.3 words favoring the non-vulnerable group), but not in the color naming task (*p* = 0.506, *d* = 0.09) despite the fact that both are considered to be measures of automatic processes. While the former task involves reading, the latter task does not, requiring participants to simply recognize and name colors. Consequently, based on these results, we would recommend the use of alternative tasks that avoid word and number reading to test cognitive inhibition in vulnerable pediatric population. One possible alternative is the inhibition task included in the NEPSY-II battery [88].

Similarly, in the TMT series, children should be able to complete this task by the age of 6, but some from our vulnerable sample were not able to. In such cases, children did not have the number system internalized and, in others, the alphabet. Given that these tasks involve the tracing of number and letter forms, it is plausible that the higher response times in both the TMT series in our vulnerable group was due to the fact that some children in this group were less familiar with the stimuli they were meant to produce compared to their counterparts in the non-vulnerable group. Although the differences found between groups in the two TMT tasks are large effect sizes (TMT-A, *d* = 1.03; TMT-B, *d* = 0.75). Nonetheless, due to the limitations previously described, these results should be interpreted with caution. In the M-WCST subtest, something similar to the Stroop results occurred as we expected to find significantly lower scores in the vulnerable group. However, somewhat surprisingly, the vulnerable group significantly outperformed the non-vulnerable group in the problem-solving task and differences in perseverative errors were non-significant. At this point, it is important to take into consideration the high rates of bilingualism found in our vulnerable group. There are studies that have analyzed the implications of bilingualism in low-SES children core processes related with the executive functions such as inhibition and shifting [89], or working memory tasks [90], among others. Bilingualism helps to enhance some cognitive processes independent of the children’s level of SES [91]. This leads us to think about the possible benefit of speaking more than one language for children that live in disadvantaged conditions. Studying a sample of 6-year-old children and taking into account both SES and the bilingual condition of the participants, Calvo and Bialystok [92] found that although children from middle-class backgrounds obtained better scores than lower-SES counterparts, the bilinguals outperformed monolinguals in executive function tasks but not in language-related tasks, independent of the SES level. These authors concluded that the influence of bilingualism and SES are both significant as well as independent predictors of children’s cognitive functioning. Additionally, in a sample of Latino low-SES preschoolers, White and Greenfield [93] compared bilinguals (Spanish-English) and monolinguals, finding that bilinguals performed better than their peers in executive function tasks. Similar conclusions have been obtained in a study with adolescents [94]. All of these results lead us to argue that bilingual conditions could compensate the adverse effects that living in vulnerable conditions causes in the children and adolescents assessed.

An important point to note about the present study is that we did not set out to compare high-SES with low-SES participants. However, we would expect to find significant differences in the vast majority of cognitive tasks if such a comparison was done. Even though the sample differs in terms of violence exposure and the grade of socioeconomic difficulties, the participants’ cognitive performance has shown to be especially different in those cognitive tasks that require cognitive processes to be more affected by extreme low-SES. Moreover, the urban educational center included in the vulnerable group is located on the outskirts of Guatemala City in an area which borders two rival criminal gangs. Consequently, the children and adolescents who attend this school were exposed to community violence on a daily basis. Future studies should consider assessing this factor as children in our non-vulnerable sample may also have been experiencing exposure to violence.

These findings, one of the first obtained in the Guatemalan pediatric population, can be a key factor to keep in mind when designing cognitive assessment protocols and intervention programs for children and adolescents who live in similar vulnerable settings. Considering the importance that attentional, linguistic, and executive processes have in academic performance, our results contribute to understanding the impact that vulnerability conditions may add in these children and adolescent cognitive development and school trajectories. This unequal situation represents a real threat for their present and future psychosocial adjustment and well-being [12], and we believe that these findings could provide relevant information for specific educational interventions and public health policies to enhance vulnerable children and adolescent cognitive development and quality of life.

### Limitations and Future Directions

Advances in this field should be directed along the lines of in-depth knowledge of how the factors associated with poverty affect cognitive development and what interventions could be effective in reducing these negative effects. Studying which cultural and socioeconomic factors associated with the vulnerability are present in both developed and developing countries may facilitate understanding of this issue. Moreover, given that cognitive development may be negatively affected by cases where children and adolescents live in vulnerability conditions, we consider it desirable to create proper adjusted scales aimed at these populations as existing scales created using only children from non-vulnerable samples are not suitable for use in these populations. The field of pediatric neuropsychology would greatly benefit from having inclusive standards that would enable the proper assessment of children and adolescents taking into account their idiosyncrasies. Regarding advice for other similar studies, we would recommend the use of the 10 neuropsychological tasks employed in this study, with the possible exception of those that rely on the influence of reading competency to assess some cognitive process. The reason for the caution regarding tests containing a linguistic component is that these types of tests could be limiting for children who, despite having a normal cognitive ability, could be unfairly assessed as exhibiting poor cognitive performance simply due to having underdeveloped reading skills (e.g., a child with an adequate inhibition ability would obtain excessively low scores in the Stroop task if they did not have an adequate reading ability).

Regarding limitations, it is important to consider that the data collected via the sociodemographic questionnaires could be influenced by social desirability. We cannot be certain that the information provided by parents regarding income and years of their own education is accurate or has been overestimated by some respondents. In relation to the executive function tests (e.g., in TMT and Stroop tests), we concluded that these results should be interpreted with caution due to the high number of children who did not complete those tasks successfully due to low reading proficiency. It is important to consider that children of the same age, but from different countries or areas, can achieve educational milestones at different developmental stages. As explained above, we recommend the use of neuropsychological tasks with a minimum reading component, as it may influence the cognitive assessment in this kind of pediatric population. Another limitation concerns the number of participants, which is insufficient for an epidemiological study and may directly affect external validity. Additionally, only participants from low- and medium-low SES backgrounds were assessed in this study. Consequently, future studies should try to increase the sample size and include participants from high SES backgrounds to improve the strength of these findings.

The long duration of the assessment process could be another weak issue in this study since fatigue during the assessment process may have influenced the performance of some children. Nevertheless, we included all 10 instruments as we wanted as broad a coverage as possible and, to lessen this potential limitation, participants could take a break during the assessment whenever they felt tired. Furthermore, as the order in which the tests were administered was randomized, this ensures that possible effects due to fatigue did not systematically influence the results of any one task—had the tests been applied in a fixed order, the results of the last tests may have been less reliable. Moreover, while the inclusion of 10 tests did result in a 2 h testing session, other researchers do not necessarily have to include all of these instruments and can instead choose just the instruments required to satisfy their research goals. A further reason for including a wide variety of instruments is that researchers from poor countries may be limited in terms of access to neuropsychological tests. By including a broad selection in the present study, we increase the chances that the information presented here will be useful for other researchers. So, for example, we included two assessments of attention, the Symbol Digit Modalities Test, SDMT and the D2 Concentration Endurance Test. Both tests revealed a significant difference between the vulnerable and non-vulnerable groups. Thus, future researchers could choose to just include one of these tests based on availability and the specific characteristics of their research. Finally, we would also like to highlight the relevance of longitudinal studies to increase our knowledge about the developmental trajectory of the abilities assessed in the present study. This knowledge would enable intervention programs to be better adjusted to a children and adolescent developmental stage.

## 5. Conclusions

In this study we aimed to promote inclusive research by evaluating the cognitive performance of a profile of children usually forgotten about in neuropsychological assessment, specifically, children and adolescents living in vulnerable conditions. To the best of our knowledge, few previous studies have focused on vulnerable populations, and this is especially true of Guatemala, which has a large pediatric indigenous population. To date, pediatric neuropsychology has scarcely represented ethnic and vulnerable minorities when assessing normative cognitive development. The sample assessed in this study was characterized for presenting different types of vulnerability such as exposure to violence and extreme poverty. These are our main conclusions:Vulnerable children and adolescents underperformed in attention, language, and some subtests of executive functions tasks, in comparison with non-vulnerable participants;Effect sizes were considerably large in the case of the PPVT-III language task and both forms of TMT executive function task, suggesting the need for specific normative data designed for children and adolescents from these backgrounds;There were no significant differences between vulnerable and non-vulnerable participants in terms of performance in motor skills and memory tasks;Bilingualism of a high percentage of the vulnerable sample may have played an important role in the obtained results, especially in executive functions’ tasks;Vulnerability may not be a risk factor for the development of all cognitive domains, since vulnerable participants obtained equal or betters scores in some subtests;In future studies, it would be important to consider the reading ability of participants and to further study how bilingualism may interfere in the cognitive assessment of vulnerable populations.

These conclusions should be considered when creating intervention programs that aim to diminish the impact that vulnerability causes in children and adolescent lives. Knowing which cognitive functions are more affected by vulnerability might favor the creation and implementation of public policies related with education that work in the advantage of millions of girls and boys from both developing and developed countries.

## Figures and Tables

**Table 1 children-08-00090-t001:** Motor skills and memory tests raw scores and independent samples *t*-test results.

	Vulnerable		Non-Vulnerable						
Tasks	*n*	*M (SD)*	*n*	*M (SD)*	*df*	*t* Statistic	CI	*p*	*d*
ROCF									
Copy	143	29.2 (8.3)	201	30.1 (6.2)	249.5	−1.1	[2.5, 0.6]	0.289	0.13
Memory	143	18.2 (9.8)	201	18.7 (7.2)	245.5	−0.5	[−2.3, 1.4]	0.587	0.06
TAMV-I									
Verbal learning	144	30.6 (8.0)	203	31.7 (7.3)	345	−1.4	[−3.0, 0.5]	0.189	0.15
Memory delay	144	8.8 (2.7)	203	9.1 (2.1)	260.2	−1.2	[−0.8, 0.2]	0.207	0.13
Recognition	144	11.1 (1.4)	203	11.2 (1.3)	345	−1.1	[−0.4, 0.1]	0.276	0.07

ROCF: Rey-Osterrieth Complex Figure Test; TAMV-I: Learning and Verbal Memory Test.

**Table 2 children-08-00090-t002:** Attention tests raw scores and independent samples *t*-test results.

	Vulnerable		Non-Vulnerable						
Tasks	*n*	*M (SD)*	*n*	*M (SD)*	*df*	*t* Statistic	CI	*p*	*d*
SDMT	137	30.3 (14.8)	203	34.3 (11.4)	241.12	−2.6	[−6.9, −0.9]	0.012	0.31
D2-CON	137	93.3 (52.7)	193	121.3 (40.7)	244.49	−5.2	[−38.1, −17.1]	0.001	0.61

SDMT: Symbol Digit Test; D2-CON: Concentration Endurance Test (D2) Concentration Index.

**Table 3 children-08-00090-t003:** Language tests raw scores and independent samples *t*-test results.

	Vulnerable		Non-Vulnerable						
Tasks	*n*	*M (SD)*	*n*	*M (SD)*	*df*	*t* Statistic	CI	*p*	*d*
PPVT-III	143	85.9 (30.0)	199	109.2 (24.6)	267.3	−7.6	[−29.2, −17.3]	0.001	0.87
Token	144	29 (3.8)	203	31 (2.7)	238.4	−5.4	[−2.7, −1.3]	0.001	0.63
Verbal fluency									
/f/	143	5.7 (4.1)	203	7.1 (3.3)	261.1	−3.2	[−2.1, −0.5]	0.002	0.38
/s/	143	6.9 (4.0)	203	7.7 (3.4)	274.3	−2.0	[−1.6, 0.0]	0.048	0.22
/a/	143	7.0 (3.9)	203	7.8 (3.3)	278.2	−2.0	[−1.6, 0.0]	0.051	0.23
Animals	144	13.7 (4.8)	203	15(4.4)	345	−2.5	[−2.3, −0.3]	0.016	0.29
Fruits	144	10.8 (3.2)	203	10.7 (2.9)	345	0.1	[−0.7, 0.7]	0.922	0.03

PPVT-III: Peabody Picture Vocabulary Test, Third edition; Token: Shortened version of the Token Test.

**Table 4 children-08-00090-t004:** Executive functions tests raw scores and independent samples *t*-test results.

	Vulnerable		Non-Vulnerable						
Tasks	*n*	*M (SD)*	*n*	*M (SD)*	*df*	*t*	CI	*p*	*d*
Stroop									
Word	114	70.1 (23.7)	190	74.4 (16.6)	180.2	−1.7	[−9.2, 0.4]	0.089	0.22
Color	114	53.6 (14.2)	190	52.4 (12.5)	214.9	0.7	[−2.1, 4.3]	0.506	0.09
Word-color	114	31.5 (8)	190	29 (9.3)	302	2.4	[0.4, 4.5]	0.014	0.28
Interference	114	1.7 (7.5)	190	−1.5 (7.1)	302	3.6	[1.6, 4.9]	0.001	0.44
TMT									
A	139	48 (22.6)	203	30.7 (11.5)	186.9	8.3	[13.5, 21.6]	0.001	1.03
B	120	80.2 (36.8)	203	57.9 (24.9)	184.4	5.9	[15.3, 29.3]	0.001	0.75
M-WCST									
Correct cat.	143	4.6 (1.8)	202	4 (1.8)	343	3.0	[0.2, 0.9]	0.006	0.33
Pers. errors	143	4.3 (4.8)	202	4.2 (3.8)	343	0.3	[−0.8, 1.2]	0.765	0.02
Total errors	143	12.9 (10.6)	202	15.2 (9.2)	343	−2.2	[−4.4, −0.2]	0.029	0.24

Stroop: Stroop Color-word interference test; TMT: Trail Making Test; M-WCST: Modified Wisconsin Card Sorting Test; Correct cat.: Correct categories; Pers. Errors.: Perseverative errors.

**Table 5 children-08-00090-t005:** Raw scores by age group and vulnerability condition.

	Children (6–11 Years)				Adolescents (12–17 Years)			
Tasks	Vulnerable		Non-Vulnerable		Vulnerable		Non-Vulnerable	
	*n*	*M (SD)*	*n*	*M (SD)*	*n*	*M (SD)*	*n*	*M (SD)*
ROCF								
Copy	86	25.33 (8.69)	128	28.60 (6.69)	57	35.09 (1.29)	73	32.76 (4.09)
Memory	86	13.67 (8.70)	128	16.26 (6.41)	57	25.13 (6.83)	73	23.09 (6.27)
TAMV-I								
Verbal learning	87	28.32 (8.32)	129	30.54 (7.71)	57	33.95 (6.14)	74	33.81 (6.16)
Memory delay	87	7.76 (2.79)	129	8.78 (2.23)	57	10.33 (1.39)	74	9.66 (1.66)
Recognition	87	10.64 (1.63)	129	11.01 (1.42)	57	11.70 (0.53)	74	11.58 (0.89)
SDMT	80	21 (10.58)	129	28.94 (9.29)	57	43.37 (8.66)	74	43.59 (8.39)
D2- CON	80	62.59 (36.65)	122	105.26 (34.44)	57	136.39 (40.20)	71	148.96 (35.65)
PPVT-III	86	69.81 (22.76)	128	99.01 (21.28)	57	110.07 (22.47)	71	127.51 (18.90)
Token	87	27.70 (4.01)	129	30.34 (2.69)	57	31.05 (2.41)	74	32.21 (2.17)
Verbal fluency								
/f/	86	3.64 (3.04)	129	6.09 (2.88)	57	8.89 (3.51)	74	8.77 (3.30)
/s/	86	5.15 (3.24)	129	6.77 (3.03)	57	9.58 (3.6)	74	9.45 (3.43)
/a/	86	5.48 (3.45)	129	6.78 (2.93)	57	9.37 (3.16)	74	9.64 (3.24)
Animals	87	11.76 (4.11)	129	13.29 (3.55)	57	16.68 (4.11)	74	17.91 (4.16)
Fruits	87	9.36 (3.14)	129	9.82 (2.41)	57	12.88 (1.98)	74	12.27 (2.98)
Stroop								
Word	57	54.42 (21.95)	120	68.59 (15.64)	57	85.82 (12.04)	70	84.34 (13.14)
Color	57	44.32 (10.53)	120	48.61 (11.27)	57	62.82 (11.11)	70	58.97 (11.96)
Word-color	57	27.14 (6.87)	120	26.32 (8.14)	57	35.88 (6.63)	70	33.63 (9.36)
Interference	57	3.41 (7.63)	120	−1.88 (7.22)	57	−0.97 (6.99)	70	−0.78 (6.88)
TMT								
A	82	51.98 (24.84)	129	32.47 (11.25)	57	42.30 (17.56)	74	27.49 (11.18)
B	63	81.80 (40.56)	129	63.13 (25.63)	57	78.41 (32.33)	74	48.72 (20.79)
M-WCST								
Correct cat.	86	4.06 (1.88)	128	3.61 (1.71)	57	5.44 (1.13)	74	4.76 (1.69)
Pers. errors	86	5.83 (5.46)	128	4.98 (3.99)	57	2.07 (2.30)	74	2.78 (2.95)
Total errors	86	16.29 (11.14)	128	17.59 (8.73)	57	7.84 (7.23)	74	11.18 (8.65)

ROCF: Rey-Osterrieth Complex Figure Test; TAMV-I: Learning and Verbal Memory Test; SDMT: Symbol Digit Test; D2-CON: Concentration Endurance Test (D2) Concentration Index; PPVT-III: Peabody Picture Vocabulary Test, Third edition; Token: Shortened version of the Token Test; Stroop: Stroop Color-word interference test; TMT: Trail Making Test; M-WCST: Modified Wisconsin Card Sorting Test; Correct cat.: Correct categories; Pers. Errors.: Perseverative errors.

## Data Availability

The data presented in this study are available on request from the corresponding author. The data are not publicly available due to the sensitivity of data collected from underage participants.

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
