# Peer review of "How Living in Vulnerable Conditions Undermines Cognitive Development: Evidence from the Pediatric Population of Guatemala"

_children, 2021, doi:10.3390/children8020090_

Round 1

Reviewer 1 Report

The study is important and compelling. 

The paper would benefit from a discussion why 10 neuropsychiatric tasks were needed, which required an assessment period of 120 minutes for the children. Why were these specific tests administered? Did the results of the tests affirm one another? At times, it was not clear whether the study centered on the comparison of children who were vulnerable and not vulnerable or the study focused on the psychometric evaluation of the 10 neuropsychiatric tests to determine pediatric standards of cognitive performance that takes environmental vulnerable conditions into consideration. Would the authors recommend the administration of all 10 tasks in the future? Would the authors recommend future assessments that take 120 minutes to administer to children?

The description of the participants (line 222) requires clarification and expansion. Are the children "vulnerable" due to extreme poverty, exposure to violence, bilingualism, parental education, indigenous status, ethnic group, or other factors (for example, rural setting vs. city)? In the description of the participants, it would be helpful to include the paragraph that begins on line 584 and the paragraph that begins on line 568. Also, would the children classified as "non-vulnerable" in the Guatemalan context be considered as "vulnerable" in a country with a larger economy (see line 188)?

The tables compare the scores of vulnerable and non-vulnerable children. Given the range of ages among the children (line 232), a table that presents the results by vulnerability status and age is also necessary. 

The description of the study mentioned on line 214 is unclear. 

Line 443 uses the term "cohorts" but the children were not followed over time. Better to use the term "groups." 

Reviewer 2 Report

Thank you for giving me the opportunity to review this paper. I congratulate the authors for their extensive work and especially for exploring a sensitive and timely research topic that definitely needs more attention.

However, the entire manuscript needs to be revised by a native English speaker. There are many spelling and semantic errors that need to be addressed by a professional - e.g.:

  • Childhood poverty is not only a reality in developing countries
  • public health policies with which will enhance vulnerable children
  • objectives such as eradicate poverty
  • children at 6 years of age
  • the idea is to ensure sufficient economic and social resources are made available to fulfill
  • Because of the implications that living in vulnerable...
  • & many more
  • were significantly worse - ?? (line 124)
  • These authors suggested that the results were due to the effects that stress causes in memory systems
  • in the social dimension there are huge gaps in the indicators of..

Also, many ambiguous paragraphs and statements need revision (e.g., To achieve the SGDs, a shift in academic research is needed (5) in order to
identify critical pathways to success based on sound research able to inform a whole new set of policies and interventions)... & many more

  • Given the heterogeneity and complexity of disadvantaged contexts, researchers have analyzed the ways in which a lack of resources affects children and adolescents from different perspectives. - Who, exactly?
  • Many authors have described how stressors caused by poverty could affect children and adolescents’ brain development (9) - needs further exploration/citations are needed
  • Child brain plasticity plays a core role in the maturation process and also makes the brain sensitive to contextual hazards. - needs further exploration / citations are needed
  • Because of the implications that living in vulnerable contexts has for children’s development and well-being, attention, perception, memory, praxias, language and executive functions of  vulnerable population have been widely studied in high-income countries - where, exactly? By who?
  • This capacity is important in early ages and is related with school
    performance, since without a proper control of this kind of attention, children could face problems with following teacher commands, and completing learning activities in which they have to attend to a target stimulus while simultaneously ignoring other irrelevant items - citations are needed. 
  • Taking this into account, results of research focusing on attention suggest that economically disadvantaged
    children have greater problems in inhibiting irrelevant information in comparison with children from
     wealthier backgrounds. - whose research?
  • Previous studies suggest that, low SES affects memory-related processes such as visuoverbal - ??
  • Nevertheless, in this study higher socioeconomic indicators were related with better performances on working memory tasks. - in which study? the present paper? the work previously cited? which one? (line 127)
  • In the same study, these authors found that only scores in immediate recall in visual memory tasks are better in medium SES children. - ? What are you trying to say, exactly?

Why are there two different sections related to Measures? (i.e., Instruments and Questionnaires)? 

How and why did you choose those specific inclusion criteria?

Cronbach’s α -s are not reported for all instruments. 

Overall, the results are interesting and bring important insights. However, the Introduction needs a lot of work, and the whole paper lacks clarity. I would recommend a more systematic approach, and a reevaluation of the sources provided to establish a theoretical frame. In the Introduction section, for example, there are many unrelated details that need to be excluded. The authors should only discuss variables that are actually relevant to their paper. 

Also, the manuscript seems excessively long and hard to follow, especially in the Procedure/Measures section. Maybe an Appendix would be a better idea.

Round 2

Reviewer 2 Report

Thank you for reviewing the paper as requested. The authors have responded to most of the concerns I've raised, though I further insist with using an Appendix as a way to shorten the manuscript, thus, improve its clarity.
